# Pulp Enhancement of Oil Palm Empty Fruit Bunches (OPEFBs) via Biobleaching by Using Xylano-Pectinolytic Enzymes of *Bacillus amyloliquefaciens* ADI2

**DOI:** 10.3390/molecules26144279

**Published:** 2021-07-14

**Authors:** Muhammad Hariadi Nawawi, Rosfarizan Mohamad, Paridah Md Tahir, Ainun Zuriyati Asa’ari, Wan Zuhainis Saad

**Affiliations:** 1Department of Microbiology, Faculty of Biotechnology and Biomolecular Sciences, Universiti Putra Malaysia, Serdang 43400, Malaysia; muhammadhariadinawawi@yahoo.com; 2Department of Bioprocess Technology, Faculty of Biotechnology and Biomolecular Sciences, Universiti Putra Malaysia, Serdang 43400, Malaysia; farizan@upm.edu.my; 3Institute of Tropical Forests and Forestry Products, Universiti Putra Malaysia, Serdang 43400, Malaysia; parida@upm.edu.my (P.M.T.); ainunzuriyati@upm.edu.my (A.Z.A.)

**Keywords:** xylanase, pectinase, *Bacillus amyloliquefaciens*, pulp and paper, oil palm empty fruit bunch (OPEFB)

## Abstract

The present work reports the biobleaching effect on OPEFB pulp upon utilisation of extracellular xylano-pectinolytic enzymes simultaneously yielded from *Bacillus amyloliquefaciens* ADI2. The impacts of different doses, retention times, pH, and temperatures required for the pulp biobleaching process were delineated accordingly. Here, the OPEFB pulp was subjected to pre-treatment with xylano-pectinolytic enzymes generated from the same alkalo-thermotolerant isolate that yielded those of higher quality. Remarkable enhanced outcomes were observed across varying pulp attributes: for example, enzyme-treated pulp treated to chemical bleaching sequence generated improved brightness of 11.25%. This resulted in 11.25% of less chlorine or chemical consumption required for obtaining pulp with optical attributes identical to those generated via typical chemical bleaching processes. Ultimately, the reduced consumption of chlorine would minimise the organochlorine compounds found in an effluent, resulting in a lowered environmental effect of paper-making processes overall as a consequence. This will undoubtedly facilitate such environmentally-friendly technology incorporation in the paper pulp industry of today.

## 1. Introduction

To date, many pulp and paper industries in the Asian countries are still employing elemental chlorine (Cl_2_) in their production processes [1], whereby a majority of pulp and paper mills worldwide now opt for Elemental Chlorine Free (ECF; ClO_2_) still throughout the bleaching phase [2]. Despite the availability of alternative eco-friendly bleaching solutions for pulp mills instead of Cl_2_ and ClO_2_ such as prolonged cooking (i.e., oxygenation) and hydrogen peroxide or ozone-based delignification, their introduction is a large-scale and cost-intensive proposal. This is attributable to the need for extensive operation adjustments or improvements, among others.

In contrast, enzymes offer a safer and highly cost-effectual approach for minimising ClO_2_, chlorine derivatives, and other chemicals for bleaching, where in contrast conventional bleaching is usually performed using harmful substances such as chlorine or bleach. However, biobleaching is the process that pulp can be bleached using an enzyme or ligninolytic microorganisms that requires minimal chemical bleach to obtain an equal pulp brightness in comparison to chemical bleach. Biobleaching is denoted as a biological technique; enzymes often propose a simple approach enabling the attainment of higher brightness ceiling, especially in the pulp and paper field. This can be achieved without necessitating a large investment in resources or expenditure alike [3]. Accordingly, xylanase and pectinase enzymes sourced from microorganisms have attracted considerable attention due to the biotechnological possibilities offered throughout various industrial processes [4,5]. Typically viewed as biocatalysts in the pulp and paper industry, they facilitate the landscape further due to the consistently shifting climate of manufacturing technologies after rising calls for improved economy, pulp quality, and environmental safety. 

However, an understanding of enzymatic pre-bleaching mechanism facilitation is currently limited. In particular, xylanase and pectinase are known as hemicellulolytic enzymes tasked with depolymerisation of hemicelluloses such as xylan and pectin, precipitated on a fibre surface. A well-known school of thought proposes enzyme incorporation via pulp structure opening and enhanced chlorine and alternative bleaching chemical access [6]. Thus, the enzyme aid and facilitate lignin removal from the pulp and boost the bleaching effect, thus enhancing the paper brightness. 

To date, enzymatic pre-bleaching is reportedly an environmentally-friendly practice [7,8,9], and it has proven its effectiveness in reducing the amount of hazardous chlorinated organic compounds such as organochlorines in bleach effluent in comparison to conventional chemical methods, thereby minimising environmental pollution. As such, reports detailing the bleaching processes of various woody and non-woody pulps by using xylanases can be identified in the literature [8,9,10,11]. Contrary to this, only a few of such reports describe the biobleaching of these pulps by incorporating pectinases [12,13], the *Bacillus pumilus*-derived production of xylanases [14], and pectinases [15,16]. Besides, publications detailing the combined usage of both aforementioned enzymes in pulp pre-treatment for biobleaching purposes are present, but the enzymes implemented are other bacterial species [17,18]. Similarly, the concurrent yield of xylanase and pectinase derived from the same strain of *B. pumilus* has been reported about the biobleaching of kraft pulp [9]. However, a description of their simultaneous generation using the identical strain of *B. amyloliquefaciens* is presently absent, as well as the efficacy of xylanase and pectinase pre-treatment on OPEFB pulp. 

Hence, the present study reports the biobleaching effect on OPEFB pulp upon implementing extracellular xylano-pectinolytic enzymes simultaneously generated from *B. amyloliquefaciens* ADI2. It is driven by the specific objective of assessing the effect of different doses, retention times, pH, and temperature required for the biobleaching process.

## 2. Results and Discussion

### 2.1. Isolation and Screening of Xylano-Pectinolytic-Producing Bacteria

Sampling from the garden soil following the property of bacterial enzyme inducibility by the substrate available in an environment was chosen as the preferred approach. Here, lignocellulosic materials are deposited, thereby offering a higher possibility of good xylanolytic and pectinolytic enzyme producer isolation [19]. In particular, one isolate was found to produce a clear zone around colonies against a dark brown on respective xylan and pectin agar upon incubation, yielding the highest clear zone diameter of 30 mm and 22 mm, respectively. It was grown for 168 h, whereby a 24-h interval for enzyme collection was employed in a liquid medium to study the isolate growth profile. As a result, 72 h of incubation time was denoted as the optimum condition for simultaneous production of xylan and pectin. The isolate was thus classified as a good producer for their simultaneous production at a ratio of 1:1.3 for xylanase-pectinase enzyme production.

### 2.2. Identification Using 16S rRNA Gene Sequencing and Phylogenetic Analysis

The bacteria’s DNA sequence consisted of 1379 bp, and was blasted in National Center for Biotechnology Information (NCBI) database, showing significant alignments of highest similarities to *Bacillus* sp. with 99% similarity to *B. amyloliquefaciens* ZN-S2 and under a similar group with ATCC strain such *B. amyloliquefaciens* DSM7 strain ATCC 23350. According to the DNA sequence, this isolated bacteria were phylogenetically related to the member of *B. amyloliquefaciens* culture collection reference strains available in GenBank from the NCBI database. Thus, the isolate was identified as *B. amyloliquefaciens* ADI2 and deposited in the GenBank database under its accession number (i.e., MG726535). Accordingly, the phylogenetic tree displayed in Figure 1 represents the evolutionary relationship in *Bacillus* species. Here, *B. amyloliquefaciens* ADI2 fell into the cluster within the *B. amyloliquefaciens* group at a bootstrap value of 1000, suggesting a shared and common ancestor.

### 2.3. Optimisation of Enzymatic Pre-Treatment Processes

The Kappa number is an efficient analytical technique for measuring the amount or degree of lignin in a completed or in-process pulp sample. The determination of reducing sugar (λ 540) is compulsory to support this evidence, as the reducing sugars were liberated from the polysaccharides (xylan and pectin) when the pulp was treated with xylano-pectinolytic enzymes, resulting in the high free sugar concentration (xylose and galacturonic acid) in the pulp sample. Furthermore, the reduction in kappa number, release of reducing sugars (λ 540), release of lignin (λ 280), and hydrophobic compounds (λ 465), coupled with aromatic compounds (λ 237 nm) during pulp bleaching are interrelated phenomena and reflect the dissociation of lignin-carbohydrate complex, and these can be measured using absorbance for rapid screening as suggested by many researchers [20,21,22,23].

Cellulase-free xylanase and pectinase from *B. amyloliquefaciens* ADI2 were grown using inexpensive lignocellulosic material (i.e., banana peel), whereby the carbon source exhibited good xylanase-pectinase production at a ratio of 1:1.3. The same medium would extensively reduce the production cost as opposed to that of individual enzymes from different microorganisms by utilizing the bacterium to produce xylanase and pectinase concurrently within. As such, the technology would be cost-effective and commercially viable alike. The selection of *B. amyloliquefaciens* ADI2 for the biobleaching pre-treatment of OPEFB pulp was supplemented by its properties, specifical stability throughout wide-ranging pH and temperature. To this end, biobleaching with the enzymatic pre-treatment revealed the optimum xylanase-pectinase dose of 15 U/g and 19.5 U/g, respectively, for the oven-dried pulp as the most effectual enzyme dosage pre-treatment (Table 1).

In contrast, pulp treated with enzyme doses is higher than the specified values failed to yield enhanced pre-treatment effectiveness. However, reports have indicated that the enzyme dosage of 5 U/g for xylanase and pectinase alike in the process of mixed hardwood and bamboo kraft pulp biobleaching [17,18].

The second experimental design for the effect of retention time was carried out, whereby 180 min was deemed fit for the pre-treatment (Table 2); beyond the aforementioned values, there was no significant improvement in biobleaching efficiency. The enzyme dosage is interrelated to the retention time. In fact, the same bleach-boosting effect can be obtained in a shorter time by increasing enzyme dosage [24]. The same phenomenon can be observed at the third experimental design for the effect of pH on pulp biobleaching, where pH 8.5 is the best condition for biobleaching pre-treatment (Table 3). Indicated xylano-pectinolytic enzymes work best under moderate alkaline conditions. The increased amount of positively-charged side chains due to alkaline conditions on the xylano-pectinolytic enzyme’s structure might have contributed to this finding [25]. As for the fourth experimental design, the effect of temperature as shown in Table 4, the optimum temperature was found suitable for pretreatment at 40 °C. The higher or lower temperature of incubation did not improve the biobleaching benefits significantly, suspected due to the thermal denaturation of enzymes structure or low kinetic energy of enzymes [26,27]. 

When subjected to the optimum conditions, the enzymatic pre-treatment could release reducing sugars as much as 66.47 ± 17.97 mg/L g of oven-dried pulp. Following the enzymatic treatment, reducing sugars found in the pulp-free filtrate were indicative of those yielded due to degraded xylan and pectin chains in the pulp fibres. Accordingly, the hydrophobic compounds (λ 465 nm) and phenolic compounds (λ 237 nm) were recorded at an absorbance of 0.22 ± 0.01 and 0.26 ± 0.04, respectively. Similarly, the characteristic peak seen post-enzymatic treatment in the pulp-free filtrate at the wavelength of 280 nm was suggestive of lignin present in the colouring matter generated. Furthermore, the enhanced filtrate absorbance observed throughout varying wavelengths post-enzymatic treatment was attributable to degraded xylan and pectin, thereby yielding maximal lignin production, reducing sugars, and chromophores from the pulp fibres.

Following enzymatic treatment, the Kappa number for the control pulp was further minimised to 110.8 units instead of its original value of 122.78 units, yielding a decrement of 9.75% reduction. The previous report indicated a reduction of 1.5 units in kappa number by xylanase from *B. licheniformis* 77-2 [28]. This could be supported by using xylanase from *B. coagulans*, thus revealing a 5.45% decrease for the kappa number of non-woody pulp [29]. Meanwhile, earlier reports detailed the synergistic action of xylanase and pectinase yielded by two bacteria, *B. pumilus* and *B. subtilis* reduced 1.2 units in kappa number, respectively [18]. Similarly, post-enzymatic pre-treatment of pectinase from *B. subtilis* SS on mixed hardwood and bamboo kraft pulp revealed a reduction of 5.85% for its pulp kappa number [12]. In xylanase-treated wheat straw pulp (i.e., enzyme dose 10 U/g), its kappa number produced a decrement of 1.1 points [30]. Therefore, these studies demonstrated a reduced kappa number following the presence of xylanase and pectinase in biobleaching pre-treatment, subsequently resulting in an improved pulp brightness. 

In summary, parameters such as enzyme dosage (i.e., xylanase-pectinase dose of 15 U/g and 19.5 U/g, respectively, for optical density (OD) pulp), pH condition (i.e., pH 8.5), temperature (i.e., 40 °C), and retention time (180 min) denoted the variables leading to higher pulp solubilisation. They displayed well-correlated links to the reducing sugars yielded and kappa number reduction.

Treated and untreated OPEFB pulps with xylano-pectinolytic enzymes were both subjected to chemical bleaching, whereby treated pulp revealed slightly higher brightness compared to its untreated counterpart. This might be attributable to the selective xylan and pectin removal, thus facilitating lignin removal as per the decrease in kappa values [31].

### 2.4. Biobleaching Using Xylano-Pectinolytic Enzymes on OPEFB Pulp

OPEFB pulp treated with xylano-pectinolytic enzymes isolated from *B. amyloliquefaciens* ADI2 revealed an enhanced brightness by 11.25%, thereby leading to 11.25% less chlorine consumption for obtaining optical pulp attributes identical to those generated by traditional chemical bleaching (Table 5). Consequently, the process is financially feasible and environmentally-friend alike, where they save other bleaching chemical costs and reduce toxic discharge, thus simplifying and reducing expenditure for wastewater treatment generated from mill effluent. Furthermore, the enzyme can be optimized and produced in high-yield and reproduced cost-effectively through cost-effective substrates or processes [2]. The 11.25% decrement in chlorine or chemical consumption was due to xylanase and pectinase’s synergistic actions, which causes xylan and pectin found in the pulp fibre to be degraded. Besides, the actions amplified the bleaching chemical access to the lignin layer of the pulp. 

Previous reports revealed a 3.68% increment in brightness for enzyme-treated hand sheets showing less chlorine consumption in kraft pulp upon concurrent implementation of xylanase and pectinase from *B. pumilus* AJK [9]. Meanwhile, eucalyptus kraft pulp revealed chlorine savings of 8% by utilising xylanase from *Streptomyces* sp. QG-11-3 [32] as reported earlier.

The kappa number of untreated pulp subjected to bleaching was 89.52 ± 4.33 upon the final stage of bleaching via enzymatic treatment following a reduction of 27.1% from 122.78 ± 0.24 (Table 5). Alternatively, the enzyme-treated pulp depicted increased brightness at every bleaching phase. In the D_1_EpD_2_ process, approximately 35.54% of brightness increment was attained for the enzyme-treated pulp by the final stage of bleaching. This was indicative of the maximum biobleaching effect due to synergistic xylanase-pectinase action during the initial phases. Meanwhile, crude xylanase isolated from *B. licheniformis* yielded 5.0 units of brightness increment upon its incorporation for eucalyptus kraft pulp bleaching [28].

Both conventional chemical bleaching and biobleaching could improve the quality of paper compared to the control. In particular, increments were seen across various parameters for the enzyme-treated pulp, namely 30% for the tensile factor, 19.4% for the bursting factor, 20.9% for the tearing factor, and 11.25% brightness compared to conventional chemical bleaching (Table 6). Here, its tensile factor could be related to the linking capacity between fibres. In contrast, the tear factor was better than the control, thus suggesting the lack of extensive cellulose matrix degradation in the enzymatic treatment [33], as well as indicating the particular effect/reaction of xylanase and pectinase. 

Furthermore, brightness improvements perceived following enzymatic pre-treatment could be attributed to two factors; xylanase action on xylan and pectinase capabilities. The first factor denotes xylanase action on xylan precipitate found on lignin, whereby the precipitate occurs due to lowered pH by the end of the cooking phase. Thus, xylanase action causes the precipitate removal and improves the bleaching chemical accessibility to the pulp fibres. Meanwhile, lignin’s capability for formulating complexes with polysaccharides like xylan and the alkali resistance attribute for some of the bonds may render them non-hydrolysed during the kraft process [34,35]. Here, xylanase causes cleavage of bonds still present linking lignin and xylan, thereby rendering the cellulose pulp structure open and xylan fragmented, following which the fragments are extracted [36].

Secondly, pectinase can weaken the complex bond structure of lignocellulosic components found between hemicellulose-cellulose-lignin frameworks. In particular, it hydrolyses pectins present in the hemicellulose, followed by the segregation of microfibrils under alkaline conditions. This causes loosening of the framework structures, which in turn become more flexible and display higher pulp fibre porosity. The mechanism results in increased accessibility of chemical agents to the pulp fibres during chemical bleaching [37], thus aiding in removing more lignin materials and ensuring higher brightness of the paper hand sheets. 

These results indicate the enzymatic pre-bleaching’s role in facilitating increased pulp fibrillation, water retention, and bonding restoration in fibres [8,10], which is possibly attributable to harsh chemical additive effects on them. Table 5 and Table 6 display enhanced enzymatic-treated pulp physical properties compared to those subjected to chemical/conventional bleaching. Thus, this study proved the benefits of biobleaching by concurrently using enzymes such as xylanase and pectinase with chemical pre-treatment, including: enhanced pulp digestibility, improved paper quality, and the lower chemical additive amount required. The last perk would minimise the amount of polluting materials seen in the effluent.

## 3. Materials and Methods 

### 3.1. Isolation and Screening of Xylano-Pectinolytic Enzyme-Producing Bacteria

Isolation of the bacteria employed for the xylano-pectinolytic enzyme production in this study was possible using garden soil samples from Taman Layang-Layang, Selayang, Malaysia. The isolated bacteria were cultivated on agar slants at 4 °C with sub-culturing carried out every three to five weeks, while the stock cultures were preserved in cryovials and stored in 20% glycerol-supplemented broth at −20 °C for long-term preservation. Following this, qualitative screening was performed by the using plate-agar approach, whereby the basic components of the medium for the screening purpose contained (g/L) peptone, 5; yeast extract, 5; KNO_3_, 5; Mg SO_4_.7H_2_O, 0.1; KH_2_PO_4_, 1, pH 8.5. 

The method first required 1.5% of agar to be prepared, whereas carbon sources such as 2% of beechwood xylan and pectin from citrus peel were added into the medium to screen respective enzyme-producing bacteria. Wells in the respective solidified agar plates were made using a sterile borer (i.e., 10 mm diameter) to load 100 μL of bacterial suspensions for screening purposes, before they were incubated for 24 h at 30 °C. Upon incubation, the agar plates were submerged by 50 mM iodine solution and left for 15 min at 30 °C for colour development. Any excess iodine solution was poured off and the clear zone diameter observed around the colony formulated was measured. Submerged fermentation was then performed for 168 h to study the growth profile of the selected bacteria.

### 3.2. Enzymes Production

In the submerged fermentation approach applied, 2% of inoculum (i.e., 24 h old) was added into 250 mL Erlenmeyer flasks, which contained 50 mL of aforementioned basal medium. They were then subjected to shaking conditions at 200 rpm for 48 h at 30 °C, following which crude enzyme harvesting was done by centrifugation at 10,000× *g* for 20 min at 4 °C. The resulting clear supernatant obtained was next utilised for xylanase, pectinase, and cellulase assays; quantification of reducing sugars [38], and estimation of protein [39].

### 3.3. Assays for Xylanase, Pectinase, and Cellulase Enzymes

Miller’s method [38] was employed in this study for performing the xylanase, pectinase, and cellulase enzyme assays accordingly. The substrates used in quantifying the respective enzymes were beechwood xylan (1%, *w*/*v*), polygalacturonic acid (PGA; 0.5%, *w*/*v*), and carboxymethyl cellulose (1%, *w*/*v*) in glycine-NaOH buffer (0.01 M, pH 8.5).

### 3.4. Identification Using 16S rRNA Gene Sequencing and Phylogenetic Analysis

Extraction and purification of the extracted genomic DNA were carried out by using FavorPrepTM Genomic DNA Mini Kit (Favorgen Biotech Corp., Taiwan). Here, amplification of a 16S rRNA fragment was performed via i-TaqTM plus DNA polymerase by using the following universal primers: 27F (5′-AGAGTTTGATCMTGGCTCAG-3′) and 1492R (5′-TACGGTTACCTTGTTACGACTT-3′). Then, the polymerase chain reaction (PCR) fragments were confirmed via gel electrophoresis in agarose 1% (*w*/*v*) and next screened under UV light. They were subsequently transported to a private laboratory (First Base Laboratories Sdn. Bhd., Seri Kembangan, Malaysia) for sequencing, following which the sequence was compared to GenBank NCBI’s nucleotide database. The neighbour-joining method of phylogenetic tree was further developed using MEGA7, whereby the tree topology was assessed via bootstrap resampling with 1000 replicates.

### 3.5. Pulp Sample and Optimisation of Enzymatic Pre-Treatment Reaction Conditions

Unbleached OPEFB pulp employed for the present study was obtained after chemi-mechanical pulping, whereby it was sieved through a 0.30 mm mesh screen and washed accordingly. It was provided by the Institute of Tropical Forestry and Forest Products (INTROP), Universiti Putra Malaysia, Selangor, Malaysia. Accordingly, the enzymatic pre-treatment studies were performed under varying reaction conditions (i.e., enzyme dosage, reaction time, pH, and temperature) to determine the superior biobleaching parameters. In particular, their optimisation was performed using the ‘one variable at a time’ approach: first, the OPEFB pulp samples (100 g oven-dried) were subjected to treatment using different xylanase-pectinase dosages (ratio 1:1.3). Here, the xylanase and pectinase dosage ranged from 0 to 25 U/g (i.e., every 5-unit interval) and 0 to 32.5 U/g (i.e., every 6.5-unit increment), respectively, of dried pulp in transparent plastic bags. The treatment was carried out at 10% consistency, whereby it was adjusted by adding distilled water and then incubated in a water bath with intermittent kneading done. Similarly, the retention time ranging from 0 to 240 min (i.e., every 60-min interval) was also optimised for attaining the maximum enzymatic efficiency, as well as optimising various pH values (i.e., 6.5 to 10.5) and temperatures (i.e., 20 °C to 60 °C). Buffer-treated pulp was utilised as the control, whereby it was subjected to the same treatment conditions without any enzyme addition [9].

### 3.6. Pulp-Free Filtrate and Pulp Sample Assessment

Enzyme-treated pulp was filtered by using a vacuum Buchner funnel to separate the residual pulp and pulp-free filtrate. In particular, the residual pulp was subjected to washing using distilled water and then air-dried in a room, before it was stored in sealed plastic bags for further processes, such as Kappa number determination and hand sheet preparation. Meanwhile, pulp-free filtrate collected was used to quantify the amount of reducing sugars released by the dinitrosalicyclic acid (DNS) method [38]. To evaluate the release of hydrophobic/phenolic compounds, an aliquot of the pulp-free filtrate was analysed using a spectrophotometer at λ 465 nm [20]. In contrast, an assessment at λ 237 nm allowed the quantification of aromatic compounds released from the lignin utilisation process [21,22]. Besides, the determination of lignin released in the filtrate was measured using UV spectroscopy at λ 280 nm [23]. Following this, the residual treated-pulp was employed to prepare hand sheets following the standard recommendations of the Technical Association of Pulp and Paper (TAPPI). The degree of delignification could be estimated using the Kappa number, measured according to TAPPI Method 236 om-99 via the reaction between potassium permanganate solution and moisture-free pulp samples.

### 3.7. Biobleaching and Chemical Bleaching of OPEFB Pulp

Residual OPEFB treated-pulp is treated with a cocktail of xylanase and pectinase (15 and 19.5 U/g, respectively, under optimised conditions) and an enzyme-untreated pulp (control) were subjected to the chemical bleaching process. It was performed in a multistage process according to the D_1_EpD_2_ methodology, where D_1_ means chlorine dioxide treatment and Ep is the subsequent step which is called alkali extraction and lastly followed by D_2,_ which is the same as D_1_, using chlorine dioxide. About 100 g of oven-dried pulps of treated and untreated types were subjected to chemical bleaching at a consistency of 10%. Their brightness was then analysed by preparing hand sheets (i.e., approximately 3 g of pulp) via standard TAPPI methods (TAPPI Test Methods, Atlanta, GA, TAPPI Press, 1996), following which the sheets were subjected to an analysis of different physical properties. 

### 3.8. Analysis of Different OPEFB Paper Hand Sheet Physical Properties 

Hand sheets were prepared under 65% humidity and 27 °C temperature for 2 h before they were subjected to characterization according to the TAPPI standard methods. This was done using different physical properties such as tensile factor (i.e., TAPPI Method 494 om-13), bursting factor (i.e., TAPPI Method 403 om-10), tear factor (i.e., TAPPI Method 413 om-12), and brightness (i.e., TAPPI Method 220 sp-01) corresponding to the respective protocols [40]. All experimental efforts were conducted in quintuplicates, whereby the outcomes were shown in the means of five.

### 3.9. Statistical Analysis

The study outcomes were subjected to a comparison by using one-way analysis of variance (i.e., one-way ANOVA) and various range tests in identifying dissimilarities between the measurement means at 5% (*p* < 0.05) significance level. The data were analysed using SPSS Statistics 17.0.

## 4. Conclusions

OPEFB pulp pre-treated with xylano-pectinolytic enzymes isolated from *B. amyloliquefaciens* ADI2 was found to yield pulp of higher quality than untreated pulp, whereby extensive pulp attribute enhancements were seen. Upon being subjected to a chemical bleaching sequence, they depicted an 11.25% increment in brightness, indicating less chlorine or chemical consumption for attaining identical optical pulp attributes generated via common chemical bleaching approaches. Ultimately, such reduction in chlorine consumption could facilitate fewer organochlorine compounds observed in the effluents, thereby improving the paper-making processes’ environmental effects. Similarly, this would aid the implementation of such thus environmentally-friendly technology within the industry.

## Figures and Tables

**Figure 1 molecules-26-04279-f001:**
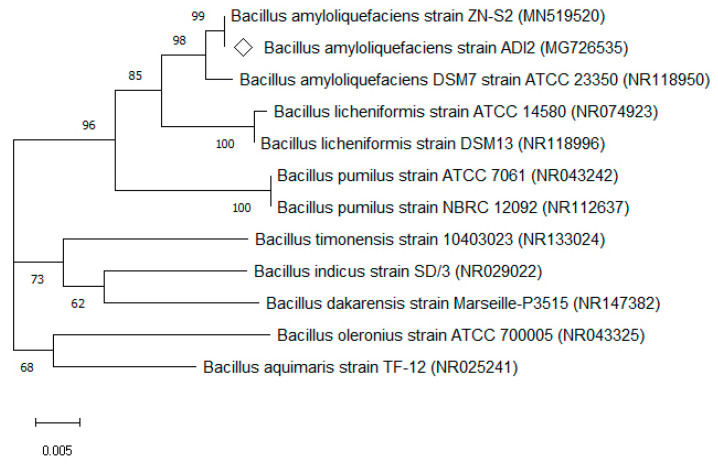
Phylogenetic correlation between *B. amyloliquefaciens* ADI2 and other *Bacillus* sp. as per the neighbour-joining approach. The bar depicts the number of nucleotide substitutions per site. The bootstrap value is attained via 1000 replications and each value is displayed at the respective nodes. The respective name and accession number of nodes are given in the tree.

**Table 1 molecules-26-04279-t001:** Effect of varying enzyme doses on pulp biobleaching.

Enzyme Dose (U/g of Oven-Dried Pulp)		Kappa No.	Reducing Sugar (mg/1 g of Pulp)	Absorbance at Wavelength (nm)		
Xylanase	Pectinase			237	280	465
0	0	122.78 ± 0.02 ^c^	4.48 ± 0.01 ^a^	4.48 ± 0.01 ^a^	0.15 ± 0.01 ^ab^	0.16 ± 0.04 ^a^
5	6.5	120.28 ± 2.34 ^c^	5.98 ± 3.42 ^a^	5.98 ± 3.42 ^ab^	0.15 ± 0.02 ^ab^	0.17 ± 0.02 ^ab^
10	13	116.52 ± 0.70 ^b^	17.18 ± 5.18 ^a^	17.18 ± 5.18 ^abc^	0.17 ± 0.02 ^b^	0.20 ± 0.02 ^bc^
15	19.5	110.80 ± 1.76 ^a^	66.47 ± 17.97 ^b^	66.47 ± 17.97 ^d^	0.20 ± 0.01 ^c^	0.22 ± 0.01 ^c^
20	26	116.73 ± 1.15 ^b^	12.70 ± 5.64 ^a^	12.70 ± 5.64 ^c^	0.15 ± 0.01 ^ab^	0.17 ± 0.02 ^ab^
25	32.5	117.22 ± 1.31 ^b^	9.70 ± 8.48 ^a^	9.70 ± 8.48 ^bc^	0.13 ± 0.01 ^a^	0.15 ± 0.01 ^a^

Experimental efforts were carried out using the following parameters: 8.5 pH, 40 °C temperature, 180 min retention time, and 10% pulp consistency. Values are given as the mean of three ± standard deviations. ^a–d^ Mean values in the same column with varying superscripts are significantly dissimilar (*p* < 0.05).

**Table 2 molecules-26-04279-t002:** Effect of retention time on pulp biobleaching.

Retention Time (min)	Kappa No.	Reducing Sugar (mg/1 g of Pulp)	Absorbance at Wavelength (nm)		
			237	280	465
0 min	122.06 ± 6.82 ^a^	2.24 ± 0.01 ^a^	0.10 ± 0.01 ^a^	0.17 ± 0.006 ^a^	0.15 ± 0.01 ^a^
60 min	120.56 ± 1.94 ^a^	10.46 ± 9.06 ^a^	0.12 ± 0.01 ^ab^	0.14 ± 0.003 ^b^	0.16 ± 0.01 ^a^
120 min	120.56 ± 1.94 ^a^	10.46 ± 7.20 ^a^	0.13 ± 0.01 ^ab^	0.16 ± 0.015 ^ac^	0.17 ± 0.02 ^a^
180 min	110.80 ± 1.76 ^b^	66.47 ± 17.97 ^b^	0.26 ± 0.04 ^c^	0.20 ± 0.011 ^d^	0.22 ± 0.01 ^b^
240 min	120.69 ± 2.43 ^a^	5.23 ± 3.42 ^a^	0.14 ± 0.01 ^b^	0.15 ± 0.01 ^bc^	0.16 ± 0.01 ^a^

Experimental efforts were carried out using the following parameters: a constant xylanase-pectinase dose of 15 and 19.5 U/g, respectively, of oven-dried pulp; 8.5 pH, 40 °C temperature, and 10% pulp consistency. Values are given as the mean of three ± standard deviations. ^a–d^ Mean values in the same column with varying superscripts are significantly dissimilar (*p* < 0.05).

**Table 3 molecules-26-04279-t003:** Effect of pH on pulp biobleaching.

pH	Kappa No.	Reducing Sugar (mg/1 g of Pulp)	Absorbance at Wavelength (nm)		
			237	280	465
pH 6.5 control	123.37 ± 3.55 ^cd^	10.46 ± 12.34 ^bc^	0.08 ± 0.02 ^b^	0.18 ± 0.06 ^ab^	0.15 ± 0.01 ^b^
pH 6.5	120.56 ± 1.94 ^bc^	25.39 ± 12.34 ^c^	0.10 ± 0.01 ^b^	0.17 ± 0.05 ^ab^	0.16 ± 0.02 ^b^
pH7.5 control	123.32 ± 1.18 ^cd^	9.71 ± 4.66 ^bc^	0.14 ± 0.03 ^c^	0.12 ± 0.01 ^a^	0.17 ± 0.01 ^b^
pH7.5	122.11 ± 0.91 ^bcd^	8.96 ± 5.93 ^bc^	0.14 ± 0.02 ^c^	0.14 ± 0.01 ^a^	0.16 ± 0.04 ^b^
pH8.5 control	122.78 ± 0.24 ^bcd^	4.48 ± 0.01 ^b^	0.09 ± 0.02 ^b^	0.15 ± 0.01 ^ab^	0.16 ± 0.04 ^b^
pH8.5	110.80 ± 1.76 ^a^	66.47 ± 17.97 ^a^	0.26 ± 0.04 ^a^	0.20 ± 0.01 ^b^	0.22 ± 0.01 ^a^
pH9.5 control	124.85 ± 2.29 ^d^	17.93 ± 10.27 ^bc^	0.14 ± 0.03 ^c^	0.15 ± 0.02 ^ab^	0.16 ± 0.01 ^b^
pH9.5	123.32 ± 1.18 ^cd^	14.94 ± 14.23 ^bc^	0.15 ± 0.01 ^c^	0.15 ± 0.01 ^ab^	0.18 ± 0.03 ^b^
pH10.5 control	124.44 ± 2.44 ^d^	11.95 ± 7.87 ^bc^	0.13 ± 0.02 ^c^	0.13 ± 0.01 ^a^	0.17 ± 0.01 ^b^
pH10.5	119.48 ± 1.31 ^b^	4.48 ± 3.88 ^b^	0.15 ± 0.01 ^c^	0.17 ± 0.01 ^ab^	0.17 ± 0.02 ^b^

Experimental efforts were carried out using the following parameters: a constant xylanase-pectinase dose of 15 and 19.5 U/g, respectively, of oven-dried pulp; 40 °C temperature, 180 min retention time, and 10% pulp consistency. Values are given as the mean of three ± standard deviations. ^a–d^ Mean values in the same column with varying superscripts are significantly dissimilar (*p* < 0.05).

**Table 4 molecules-26-04279-t004:** Effect of different temperatures on pulp biobleaching.

Temperature	Kappa No.	Reducing Sugar (mg/1 g of Pulp)	Absorbance at Wavelength (nm)		
			237	280	465
Control (25 °C)	125.61 ± 1.94 ^a^	6.722 ± 2.241 ^a^	0.150 ± 0.009 ^a^	0.15 ± 0.01 ^a^	0.17 ± 0.01 ^ab^
20 °C	123.19 ± 0.47 ^a^	11.95 ± 10.10 ^a^	0.15 ± 0.01 ^a^	0.15 ± 0.01 ^a^	0.16 ± 0.02 ^a^
30 °C	122.91 ± 2.70 ^a^	10.46 ± 3.42 ^a^	0.16 ± 0.03 ^a^	0.16 ± 0.02 ^a^	0.18 ± 0.01 ^b^
40 °C	110.80 ± 1.76 ^b^	66.47 ± 17.97 ^b^	0.26 ± 0.04 ^b^	0.20 ± 0.01 ^b^	0.22 ± 0.01 ^c^
50 °C	122.42 ± 0.88 ^a^	9.71 ± 4.66 ^a^	0.17 ± 0.01 ^a^	0.17 ± 0.01 ^a^	0.17 ± 0.01 ^ab^
60 °C	123.78 ± 5.21 ^a^	2.99 ± 1.29 ^a^	0.16 ± 0.01 ^a^	0.16 ± 0.01 ^a^	0.18 ± 0.01 ^ab^

Experimental efforts were carried out using the following parameters: a constant xylanase-pectinase dose of 15 and 19.5 U/g, respectively, of oven-dried pulp; 8.5 pH, 180 min retention time, and 10% pulp consistency. Values are given as the mean of three ± standard deviation. ^a–c^ Mean values in the same column with varying superscripts are significantly dissimilar (*p* < 0.05).

**Table 5 molecules-26-04279-t005:** A comparison of oil palm empty fruit bunch (OPEFB) pulp attributes according to chemical bleaching and enzymatic bleaching (biobleaching).

Parameters (%)	Enzyme	Chemical
Kappa no. (unbleached pulp)	122.78 ± 0.24	122.78 ± 0.24
Brightness (%ISO)	18.05 ± 0.31	18.05 ± 0.31
Dioxide D1 Stage		
Kappa no.	101.87 ± 0.58	111.21 ± 2.25
D1 added, chlorine, %	2	2
pH maintained	2.0–2.25	2.0–2.25
Brightness, %ISO	21.22 ± 0.30	20.16 ± 0.42
Extraction Stage (Ep)		
NaOH added, %	1.5	1.5
Peroxide added, %	1	1
Kappa no.	97.81 ± 2.19	103.68 ± 1.28
pH maintained	10.5–11.0	10.5–11.0
Brightness, %ISO	24.20 ± 0.91	21.58 ± 0.11
Dioxide D2 Stage		
Dioxide added, %	1.25	1.25
Kappa no.	89.52 ± 4.35	100.80 ± 1.22
pH maintained	3–4	3–4
Brightness, %ISO	28.00 ± 0.12	24.85 ± 0.51

Values are denoted as the mean of three ± standard deviation. Pulp pre-treated with xylano-pectinolytic enzyme and subjected to the subsequent D_1_EpD_2_ bleaching sequence yielded physical attributes with extensive enhancement. The paper quality was greatly improved as seen per its bursting factor, tensile strength, tearing factor, and brightness.

**Table 6 molecules-26-04279-t006:** Various physical attributes of biobleaching OPEFB pulp.

Pulp Properties	Control (Untreated and Unbleached)	Chemical (without Enzymatic Treatment)	Enzymatic Treatment
Tensile factor	0.18 ± 0.01 ^a^	0.21 ± 0.02 ^b^	0.30 ± 0.01 ^c^
Bursting factor	15.88 ± 2.01 ^a^	16.62 ± 2.02 ^a^	20.62 ± 3.07 ^b^
Tearing factor	124 ± 8.94 ^a^	136 ± 8.94 a	172 ± 10.90 ^c^
Brightness (%ISO)	17.71 ± 0.10 ^a^	24.85 ± 0.51 ^b^	28.00 ± 0.12 ^c^

Values are given as the mean of five ± standard deviations. ^a–c^ Mean values in the same row with varying superscripts are significantly dissimilar (*p* < 0.05).

## Data Availability

Data is contained within the article.

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
