# Peer review of "Pulp Enhancement of Oil Palm Empty Fruit Bunches (OPEFBs) via Biobleaching by Using Xylano-Pectinolytic Enzymes of Bacillus amyloliquefaciens ADI2"

_molecules, 2021, doi:10.3390/molecules26144279_

Round 1

Reviewer 1 Report

The production of economic and environmental pulp is an interesting topic. The manuscript is well prepared; however, the authors should address the following recommendations:

- Biobleaching could be defined in the Introduction section.

- The authors should specify why Kappa number, Reducing Sugar and Absorbance are used to evaluate the effect of various parameters (enzyme dose, time, T, pH) on biobleaching.

- What is the meaning of the letters next to the values of the parameters in Tables 1 to 4 and Table 6?

- Please see the 1st sentence in the Introduction section, there is a repetition of “to date”.

- Reference to Table 3 and Table 4 is required in the text.

Author Response

Response to Comments

Point 1:  Biobleaching could be defined in the Introduction section.

Response 1: The definition of biobleaching has been improved in Introduction section:

Line 40: “Where conventional bleaching is usually performed using harmful substances such as chlorine or bleach. However, for biobleaching, is the process that pulp can be bleached using an enzyme or ligninolytic microorganisms that requires minimal chemical bleach to obtain an equal pulp brightness as chemical bleach. Biobleaching denoted as a biological technique, enzymes often propose a simple approach enabling the attainment of higher brightness ceiling, especially in the pulp and paper field.”

Point 2: The authors should specify why Kappa number, Reducing Sugar and Absorbance are used to evaluate the effect of various parameters (enzyme dose, time, T, pH) on biobleaching.

Response 2: Reasons of Kappa number, reducing sugars and absorbance being used to evaluate the effect of various parameters are specified:

Line 111: “The Kappa number is an efficient analytical technique for measuring the amount or degree of lignin remained in a completed or in-process pulp sample. To support this evident, the determination of reducing sugar (λ 540) is compulsory, as the reducing sugars were liberated from the polysaccharides (xylan and pectin) when the pulp was treated with xylano-pectinolytic enzymes, resulting in the high free sugar concentration (xylose and galacturonic acid) in the pulp sample. Furthermore, the reduction in kappa number, release of reducing sugars (λ 540), release of lignin (λ 280), hydrophobic compounds (λ 465) coupled with aromatic compounds (λ 237nm) during pulp bleaching are interrelated phenomena and reflect the dissociation of lignin-carbohydrate complex and these can be measured using absorbance for rapid screening as suggested by many researchers [20–23].”

Line 347: “To evaluate the release of hydrophobic/phenolic compounds, an aliquot of pulp-free filtrate was analysed using a spectrophotometer at λ 465 nm [20], whereas an assessment at λ 237 nm allowed the quantification of aromatic compounds released from the lignin utilisation process [21,22]. Besides, the determination of lignin released in the filtrate was measured using UV spectroscopy at λ 280 nm [23].”

Point 3: What is the meaning of the letters next to the values of the parameters in Tables 1 to 4 and Table 6?

Response 3: The meaning of the letters next to the values of the parameters in Table 1 to 4 and 6 is the statistical analysis superscript, to find the differences between the measurement means at the 5% significant level. Where, varying superscripts are significantly dissimilar (P < 0.05).

Point 4: Please see the 1st sentence in the Introduction section, there is a repetition of “to date”.

Response 4: The 1st sentence has made the correction to this matter.

Line 31: “To date, many pulp and paper industries in the Asian countries are still employing elemental chlorine (Cl2) in their production processes [1],”

Point 5: Reference to Table 3 and Table 4 is required in the text..

Response 5: References to Table 3 and 4 have been provided.

Line 153: The same phenomenon can be observed at the third experimental design for the effect of pH on pulp biobleaching, where pH 8.5 is the best condition for biobleaching pretreatment (Table 3).

Line 157: As for the fourth experimental design, the effect of temperature as shown in Table 4, the optimum temperature was found suitable for pretreatment at 40°C.

We do sincerely appreciate the constructive comments given by reviewer. Thank you for the invaluable ideas and guidance throughout this manuscript.

With kinds regards,

Author

Reviewer 2 Report

In this study, the authors set out to isolate microorganisms with xylanase and pectinase activity from soil samples for biotechnological purposes. The main motivation was to identify a strain that would be able to improve the bleaching of oil palm empty fruit bunches for industrial purposes, but in an environmental friendly manner. The authors successfully isolated an alkalo-thermotolerant strain, identified as Bacillus amyloliquefaciens ADI2, that secreted xylanase and pectinase under culture conditions. Next, the authors determined the optimal conditions for xylano-pectinolytic activity of the culture supernatant (pH, temperature, incubation time) and compared the effect of the enzymatic preparation with that of conventional chemical bleaching on several properties of bleached pulp. The methodology is scientifically sound and conclusions are supported by data. However, there are some unsupported statements that need references or further explanation. Noteworthy, the text requires major editing of English language and style. See my specific concerns numbered below:

  1. The paper is very interesting and clearly reached its goals following technically sound procedures. However, its content is unclear, and sometimes confusing due to poor writing. At several instances, the reader is unsure about what the authors mean and odd terminology is used. Thus, the text should be revised for English quality.

  2. Line 43: This statement gives the idea that the production of enzymes is inexpensive at all, but it has its costs. To strengthen this argument, the authors should should provide an actual comparison of the costs for each approach. 
  3. Figure 1: Consider highlighting Bacillus amyloliquefaciens ADI2 so that the readers can readily recognize it.
  4. Discussion: The authors should explore and explain why the enzyme preparation is not as effective under non-optimal conditions, especially for the incubation time. In other words, why the concentration of reducing sugar and the absorbance at the three wavelengths decrease at 240 min?
  5. Line 180: Please provide evidence to support that the "process is financially feasible". 
  6. Consider moving the Conclusion to after the discussion and before the methodology. Some readers might not even read it at the current position.
  7. Include a reference for "TAPPI, 1996" in the references list.
  8. Line 349: Higher quality compared with what? Please specify. 
  9. Line 105: Use italics to refer to species.

Author Response

Response to Comments

Point 1:  The paper is very interesting and clearly reached its goals following technically sound procedures. However, its content is unclear, and sometimes confusing due to poor writing. At several instances, the reader is unsure about what the authors mean and odd terminology is used. Thus, the text should be revised for English quality..

Response 1: The manuscript is currently uploaded for language editing under MDPI Language Pre-Editing Service.

Point 2: Line 43: This statement gives the idea that the production of enzymes is inexpensive at all, but it has its costs. To strengthen this argument, the authors should should provide an actual comparison of the costs for each approach.

Response 2: The statement in Line 40 has been strengthen.

Line 40: “Where conventional bleaching is usually performed using harmful substances such as chlorine or bleach. However, for biobleaching, is the process that pulp can be bleached using an enzyme or ligninolytic microorganisms that requires minimal chemical bleach to obtain an equal pulp brightness as chemical bleach. Biobleaching denoted as a biological technique, enzymes often propose a simple approach enabling the attainment of higher brightness ceiling, especially in the pulp and paper field.”

Point 3: Figure 1: Consider highlighting Bacillus amyloliquefaciens ADI2 so that the readers can readily recognize it.

Response 3: Figure 1 has been changed, and Bacillus amyloliquefacien ADI2 has been highlighted.

Point 4: Discussion: The authors should explore and explain why the enzyme preparation is not as effective under non-optimal conditions, especially for the incubation time. In other words, why the concentration of reducing sugar and the absorbance at the three wavelengths decrease at 240 min?

Response 4: Discussion has been improved regarding this section.

Line 31: “The second experimental design for the effect of retention time was carried out, whereby 180 min was deemed fit for the pre-treatment (Table 2), beyond the aforementioned values, there was no significant improvement in biobleaching efficiency. The enzyme dosage is interrelated to the retention time. In fact, the same bleach-boosting effect can be obtained in shorter time by increasing enzyme dosage [24]. The same phenomenon can be observed at the third experimental design for the effect of pH on pulp biobleaching, where pH 8.5 is the best condition for biobleaching pretreatment (Table 3). Indicated xylano-pectinolytic enzymes work best under moderate alkaline condition. The increased amount of positively-charged side chains due to alkaline condition on the xylano-pectinolytic enzymes structure might have contributed to this finding [25]. As for the fourth experimental design, the effect of temperature as shown in Table 4, the optimum temperature was found suitable for pretreatment at 40°C. Higher or lower temperature of incubation did not improve the biobleaching benefits significantly, suspected due to the thermal denaturation of enzymes structure or low kinetic energy of enzymes [26,27].”

Point 5: Line 180: Please provide evidence to support that the "process is financially feasible".

Response 5: Evidence to support this statement has been provided.

Line 153: “Consequently, the process is financially feasible and environmentally-friend alike, where they save other bleaching chemical costs and reduction in toxic discharge, thus simplify and reduce expenditure for wastewater treatment generated from mill effluent. Furthermore, the enzyme can be optimized and produced in high-yield and reproduced cost-effectively through cost-effective substrates or processes [33].”

Point 6: Consider moving the Conclusion to after the discussion and before the methodology. Some readers might not even read it at the current position.

Response 6: The format’s flow is adapted from MDPI style, where conclusion after the methodology section.

Point 7: Include a reference for "TAPPI, 1996" in the references list.

Response 7: Reference for TAPPI has been included in the references list.

Point 8: Line 349: Higher quality compared with what? Please specify

Response 8: The sentence has been improvised.

Line 384:B. amyloliquefaciens ADI2 was found to yield pulp of higher quality compared to untreated pulp,”

Point 9: Line 105: Use italics to refer to species.

Response 9: The un-italics bacteria species has been italicised.

We do sincerely appreciate the constructive comments given by reviewer. Thank you for the invaluable ideas and guidance throughout this manuscript.

With kinds regards,

Author
